# PCNN: Environment Adaptive Model Without Finetuning

## Abstract

Convolutional Neural Networks (CNNs) have achieved tremendous success for many computer vision tasks, which shows a promising perspective of deploying CNNs on mobile platforms. An obstacle to this promising perspective is the tension between intensive resource consumption of CNNs and limited resource budget on mobile platforms. Existing works generally utilize a simpler architecture with lower accuracy for a higher energy-efficiency, *i.e.*, trading accuracy for resource consumption. An emerging opportunity to both increasing accuracy and decreasing resource consumption is **class skew**, *i.e.*, the strong temporal and spatial locality of the appearance of classes. However, it is challenging to efficiently utilize the class skew due to both the frequent switches and the huge number of class skews. Existing works use transfer learning to adapt the model towards the class skew during runtime, which consumes resource intensively. In this paper, we propose **probability layer**, an *easily-implemented and highly flexible add-on module* to adapt the model efficiently during runtime *without any fine-tuning* and achieving an *equivalent or better* performance than transfer learning. Further, both *increasing accuracy* and *decreasing resource consumption* can be achieved during runtime through the combination of probability layer and pruning methods.

## 1 Introduction

Convolutional Neural Networks (CNNs) have achieved tremendous success for many computer vision tasks, such as image classification, object detection, and image segmentation. The success on these vision tasks has fueled the desire to deploy CNNs on mobile platforms, due to the pervasive use of mobile and wearable devices, and the rise of emerging applications such as self-driving car.

While the deployment of CNNs on mobile platforms provides a fascinating perspective, two significant but conflicting challenges stand in the way of the future deployment: **accuracy** and **resource-efficiency**. The strict resource requirements on mobile platforms generally forces the deployment of a simpler model with lower accuracy, as trading accuracy for resource-efficiency. Therefore, how to improve the accuracy given the limited resource budget has become an important research topic in the filed of deep learning, and also is the focus of this paper.

An emerging opportunity to resolve the conflict between accuracy and resource-efficiency on mobile platforms is the **class skew**, *i.e.*, the strong temporal and spatial locality of the appearance of classes. For example, only a few people may appear in our lab, even if we may meet thousands of people through the whole year. While a complex model with thousands of classes is required to classify thousands of people, a small model with less than 10 classes can be sufficient in the lab. Less number of classes indicates a higher accuracy and less resource consumption. For example, if we randomly guess from 1000 classes, the accuracy is 0.1%, while the accuracy would increase to 10% for randomly guessing from 10 classes. This increase in accuracy provides more space on pruning the architecture, since the main constraint of pruning methods is the decrease in accuracy. The fascinating perspective of both *increasing accuracy* and *decreasing resource consumption* motivates the utilization of class skew on mobile platforms.

While utilizing class skews provides a fascinating perspective, it is challenging to adapt the model efficiently towards the class skew during runtime, especially when we consider the frequent switches of class skews, *e.g.*, every 10 minutes. It is also infeasible to pre-train a sequence of models for each class skew, since there is a mind-bogglingly huge number of class skews. For example, if we take

10 out of 100 classes, there would be $1.7310^{14}$ class skews. Existing works choose to fine-tune the model towards the class skew during runtime when class skew switches, based on the technique from transfer learning. With transfer learning, the number of nodes in the last layer will be reduced according to class skew and last few layers will be fine-tuned by several epochs, which introduce lots of computation overhead and latency. A 14-second or even minutes latency will occur every time the class skew switches and the model is adapted.

To efficiently adapt the model during runtime towards the class skew, we bring up **probability layer**, a novel add-on module with contributions as the following.

- No overhead is introduced while probability layer can still adapt the model towards the class skew, achieving an equivalent or better accuracy than transfer learning.

- An effective **C**lass **S**kew **D**etector (**CSD**) is proposed to efficiently detect the class skew, serving as the base for model adaption.

- Both increasing accuracy and decreasing resource consumption can be achieved by probability layer when combined with pruning techniques, which is critical to the successful deployment of CNNs on mobile platforms.

## 2 RELATED WORKS

**Environment information**  Environment information is a promising approach for energy-efficiency in the deployment of deep learning techniques. (Kang et al. (2017)) skips redundant images from surveillance cameras fixed at a crosswalk, by comparing the input stream with the fixed background. (Jiang et al. (2018)) utilizes the temporal redundancy of input stream to reduce the processing rate while keeping a similar accuracy. In this paper, we utilize another environment information, *class skew*, which is complimentary to existing approaches.

Our approach shares some similarities with the previous work (Han et al. (2016); Shen et al. (2017)) in utilizing class skew during runtime. Compared to their approach that uses transfer learning to adapt the model towards class skew, we can adapt the model without any overhead and achieving an equivalent or better performance, which suits better with the nature of limited resource budget on mobile platforms.

**Transfer learning**  Transfer learning is currently the dominant method for domain adaption problems. In transfer learning, we will have a source domain and a target domain. A model will be pretrained on the source domain and the last few layers will be retrained on the target domain such that the model can handle testing data from the target domain (Doersch et al. (2015); Han et al. (2016); Oquab et al. (2014); Shen et al. (2017); Yosinski et al. (2014)). Benefits of transfer learning have been shown in various areas, including utilizing unlabeled data (Doersch et al. (2015); Noroozi & Favaro (2016)) and reconciling the lack of data in the target domain (Oquab et al. (2014); Yosinski et al. (2014)).

While the transfer learning shows merit in domain adaption and utilized in (Han et al. (2016); Shen et al. (2017)) for runtime model adaption towards class skews, it does not fit well with the nature of limited resource budget and desired property of low latency on mobile platforms. It is also hard to conduct transfer learning without human guidance on mobile devices, due to the complex decision on how many layers we should fine-tuning, as reported by (Yosinski et al. (2014)), and the hyper-parameters utilized in the training process, *e.g.*, number of epochs and learning rate, which has important influence over the fine-tuning process.

Comparing to transfer learning, our approach fits better with the limited resource budget on mobile platforms by *requiring no fine-tuning* for model adaption while still provides an *equivalent or better* accuracy.

**Pruning methods**  Pruning methods utilize various model redundancy to decrease resource consumption with the cost of accuracy. Quantization replace convolutional filters with quantilized vectors (Bagherinezhad et al. (2017)) to reduce computation and replace 32-bit weights with 8-bit weights (Han et al. (2015)) to use less memory. Filter pruning (Lin et al. (2018)) reduces computation by decreasing number of filters that has little influence over the accuracy. Fabrics (Saxena & Verbeek

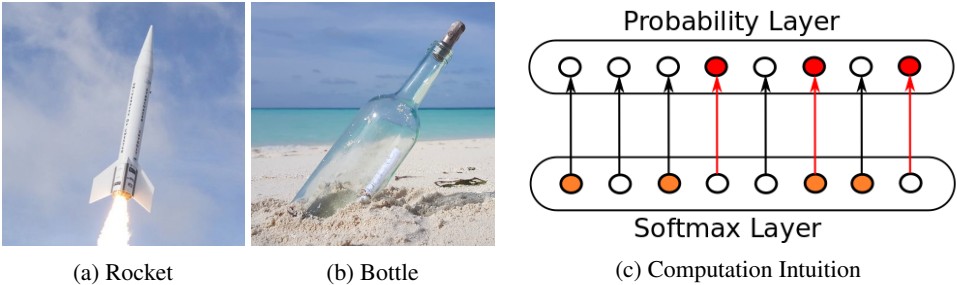

(a) Rocket        (b) Bottle        (c) Computation Intuition

Figure 1: Probability Layer Intuition. Figure 1a and Figure 1b give example of similar classes and Figure 1c shows how probability layer and softmax layer take effect together.

(2016); Huang et al. (2017a); Tom & Ludovic (2018)) targets for discovering the optimal architecture by explore an exponentially large number of architectures. While pruning methods reduce resource consumption, accuracy is also decrease. Instead, both *decreasing resource consumption* and *increasing accuracy* can be achieved during runtime without any overhead, through combining pruning methods with probability layer.

## 3 PROBABILITY LAYER

In this section, we introduce the probability layer and the PCNN. Before proceeding further, we introduce the notation that will be used in the rest of the paper.

**Notation.** CNN can be viewed as a feed-forward multi-layer architecture that maps the input images $X$ to a vector of estimated probability for each class $\vec{p} = (p_1, p_2, ..., p_n)$, where $n$ is the number of classes and $p_i = P(i|X)$ is the estimated probability $p_i$ for the label $i$ given the input image $X$.

**Key Assumption.** The main difference between the proposed layer and the original CNNs is that we take the environment information into consideration. In the original CNNs, the prediction for each image will be made individually, assuming a sequence of images is independent and identically distributed (*i.i.d*). However, in real life, this assumption does not hold and strong temporal and spatial locality may exist. Instead, we assume that class skew exists, which means the number of classes and class types are fixed during a time period. Meanwhile, it is still possible that the class skew switches to another class skew after a few minutes, which can be detected and handled by the profiler, as detailed in section 4. Further, we assume the existence of a model classifying all possible classes, called the *general model*. This assumption is reasonable because modern CNNs can be trained to classify thousands of classes (Krizhevsky et al. (2012); Simonyan & Zisserman (2014); Szegedy et al. (2015); He et al. (2016); Huang et al. (2017b)).

**Intuition.** Probability layer helps by using environment information that original CNNs does not use. When human recognizes an object, both vision and environment information will be used, i.e., what we have seen recently and which objects may appear here. However, CNNs can only make use of visual information while discarding environment information, which makes it extremely difficult to distinguish classes with similar shapes. For example, Figure 1a and 1b shows images from CIFAR-100 for bottle and rocket respectively. It is hard to distinguish these two classes only from images while environment information can easily rule out rocket in most scenarios.

Figure 1 gives intuition on how probability layer utilize environment information. In Figure 1c, the lower row represents the outputs from softmax layer and the upper row represents the probability layer. The orange nodes stand for the classes with high predicted probability in softmax layer and the red nodes stand for the suggestion from the environment. The prediction from probability layer will be selected from the intersection of the set of red nodes and orange nodes, which rules out confusing classes for CNNs.

**Approach.** Probability layer is an extra layer after the CNN model, rescaling the output of softmax layer. Rescaling is a topic in statistics (Saerens et al. (2002)). To the best of our knowledge, we are the first to discuss rescaling in CNN context. The outputs of original CNNs predict the probability

for each class and the probability layer will adjust this prediction based on the difference of class distribution in training and testing dataset. In particular, for classes with different distributions in training and testing dataset, the probability layer will rescale the corresponding outputs from softmax layer according to the difference in distribution. For other classes with the same distribution in both training and testing dataset, the outputs of the proposed layer are equal to the outputs of the softmax layer.

The probability layer will take as input the originally predicted probability, class distribution in training dataset, as well as the distribution in testing dataset, and output a vector for the rescaled prediction. The first input is the prediction vector $\mathbf{P}(\cdot|X)$ from the softmax layer, which represents the originally predicted probability for each class from the original CNNs. The second input is a vector of class distribution $\mathbf{P}(\cdot)$ in training dataset and the third one is a vector of class distribution $\mathbf{P}_t(\cdot)$ in testing dataset. The probability layer will rescale the predicted probability in $\mathbf{P}(\cdot|X)$ element wisely and produce as output a vector $\mathbf{P}_t(\cdot|X)$ for the rescaled prediction of each class.

Formally, let the outputs of CNNs with and without rescaling are

$$P_t(i|X) = \frac{P_t(X|i) \cdot P_t(i)}{P_t(X)} \tag{1}$$

and

$$P(i|X) = \frac{P(X|i) \cdot P(i)}{P(X)} \tag{2}$$

respectively. Here $P_t(i)$ means the class distribution in testing dataset and $P_t(i|X)$ represents the predicted probability for class $i$ after the probability layer. We assume that $P_t(X|i)$ equals $P(X|i)$ approximately, where $P(X|i)$ is the distribution of image data for class $i$. This assumption makes sense since, for a class $i$, the selection of input x is random. Through transforming equation 4 and equation 5 as well as utilizing $P_t(X|i) = P(X|i)$, we can derive that

$$P_t(i|X) = \frac{P_t(i)}{P(i)} \cdot \frac{P(X)}{P_t(X)} \cdot P(i|X). \tag{3}$$

Considering $\sum_{i=1}^{n} P_t(i|X) = 1$, we can get the rescaling formular as

$$P_t(i|X) = \frac{\frac{P_t(i)}{P(i)} \cdot P(i|X)}{\sum_{j=1}^{n} \frac{P_t(i)}{P(j)} \cdot P(j|X)} \tag{4}$$

To give probability layer the ability to detect new classes, we choose not to rescale the outputs from softmax layer when the original model has strong confidence in its prediction and set the formula of probability layer as

$$P_t(i|X) = \frac{\frac{P_t(i)}{P(i)} \cdot P(i|X)}{\sum_{j=1}^{n} \frac{P_t(i)}{P(j)} \cdot P(j|X)} \cdot I_{\{P(i|X)<\omega\}} + P(i|X) \cdot I_{\{P(i|X)>=\omega\}}, \tag{5}$$

where $\omega$ is the threshold above which we should trust the original prediction and $I_X$ is the indicator function such that $I_X(x) \triangleq$ if $x \in X$, return 1, otherwise return 0. If a model has a strong confidence in its prediction, the accuracy would be much higher than the model's average accuracy. Our experiments show that CNNs will give most of the images high predicted probability and the accuracy of these images will exceed average accuracy a lot. Probability layer helps when the original model is confused on the prediction and will not interfere with the decision when the original model has confidence in its prediction.

## 4   CLASS SKEW DETECTOR

**C**lass **S**kew **D**etector (**CSD**) detects class skew efficiently, serving as the base for probability layer. We denote a stream of images to be classified as $x_1, x_2, ..., x_i, ... \in X = \mathbb{R}^n$ and the corresponding true labels as $y_1, y_2, ..., y_i, ... \in Y = [1, ..., k]$. Assume a partition $\pi : I^+ \to I^+$ over the stream exists, where each partition maintains a distribution $T_{\pi(i)}$ and the image $(x_i, y_i)$ is drawn randomly from distribution $T_{\pi(i)}$. Here, the overall series is an abruptly-changing and piece-wise stationary distribution. At test time, neither true labels $y_i$ nor partition $\pi$ is known. Also we do not have any assumptions on how long a stationary distribution exist. The task of CSD is to detect the stationary distribution when it appears and switch to another stationary distribution when it switches.

While the duration of class skew cannot be decided easily, we detect the class skew in a windowed style, which provides an approximate estimation of the duration. Supposing that every $w_{min}$ frames form a window, we can run the full model on each frame and record the distribution in these $w_{min}$ (= 30) frames. We can further compare the record in $S_j$ and $S_{j-1}$. If the difference in appearance times is less than a threshold $\pi_r$ (=2), we conclude that the previous epoch is continuing and use their concatenation as the estimation of class skew. In addition, the detection of the change in class skew can be achieved by equation 5, since the probability layer will not interfere with the decision when the original model has confidence in its prediction.

---

**Algorithm 1** CSD algorithm

**function** CSD()
  **for** $i$ in $1, ..., w_{min}$ **do**
    $y_t \leftarrow h(t)$
    $S_j \leftarrow S_j \oplus [y_t]$
  **end for**
  **if** $||S_{j-1}, S_j|| \leq \pi_r$ **then**
    $S_j \leftarrow S_{j-1} \oplus S_j$
  **end if**
  **return** $S_j$
**end function**

---

## 5 EXPERIMENTS

To show the potential of probability layer, we evaluate probability layer on specialized datasets composed of various numbers of classes and class types. We begin our experiments by showing that, with probability layer on **static class skew** with strong class skew (p=1) or weak class skew (p¡1), consistent benefit is provided *over all numbers of class skews* and, without any overhead, an *equivalent or better performance* than transfer learning can be achieved. We proceed to exhibit that, under **dynamic class skew**, *i.e.*, class skews that are stationary over a time period and switch later, combining CSD and probability layer can efficiently detect class skew and achieve a similar performance as the case under static class skew. Finally, we demonstrate that probability layer is complementary to acceleration methods through combining probability layer with spatial redundancy elimination approach (Figurnov et al. (2016)) to both decrease computation and increase accuracy.

**Dataset.** Two commonly used image classification datasets, CIFAR-100 (Krizhevsky & Hinton (2009)) and ImageNet (Deng et al. (2009)), serve as the base for generating specialized dataset. To generate a specialized dataset with $n$ classes occupying percentage $p$ in all images, we randomly select $n$ classes, each occupying percentage $\frac{p}{n}$. The other $1 - p$ percentage is occupied by images from other classes. In evaluation, a model pre-trained on the specific dataset will be tested on the specialized dataset generated from the same dataset. The preprocessing of ImageNet dataset follows the approach described in (Szegedy et al. (2016)). 1-crop result of probability layer on the validation dataset is reported in section 5.4.

**Base model.** We evaluate probability layer on five state-of-the-art architectures, *i.e.*, InceptionNet (Szegedy et al. (2015)), MobileNet (Howard et al. (2017)), VggNet (Simonyan & Zisserman (2014)), ResNet (He et al. (2016)), and DenseNet (Huang et al. (2017b)). We reimplemented DenseNet on Tensorflow (Abadi et al. (2016)) and trained the model on CIFAR-100 (Krizhevsky & Hinton (2009)) from scratch. For the other four model, we use the pretrained model on ImageNet provided by (Silberman & Guadarrama (2017)).

### 5.1 PROBABILITY LAYER ON STATIC CLASS SKEW

In this section, we show the benefit from probability layer on static class skews, *i.e.*, class skews with fixed class number and types. Two types of class skews are considered, *i.e.*, strong class skew (p=1) where a few classes occupy all the input stream, and weak class skew (p¡1), in which a few classes account for most of the input stream while other classes may still appear. Further, we compare our probability layer with the commonly used runtime model adaption method, *i.e.*, transfer learning.

**Strong class skew ($p = 1$) with different number of classes.** When the number of classes reduces, benefit can be introduced by both probability layer and transfer learning, as detailed by Figure 2. To measure the performance on a class skew with $n$ classes, we randomly sampled $n$ classes for 100 times and present the average accuracy. We see that significant benefit has been achieved by probability layer for all numbers of classes. When there are 5 classes, more than 20% increase in accuracy can be achieved without any finetuning. Another point worth noting is that the benefit

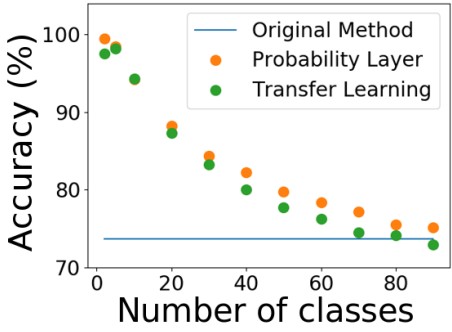

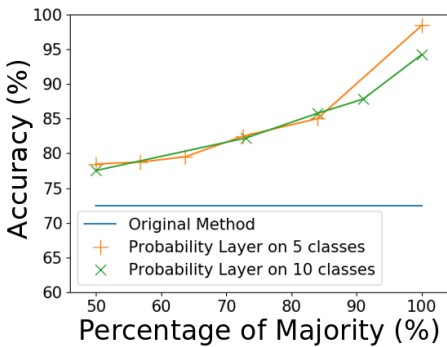

Figure 2: Strong class skew ($p = 1$) with different number of classes

Figure 3: Weak class skew ($p < 1$) with various percentage p

diminishes slowly as the number of classes increases. Even if there are $40$ classes, a benefit over $10\%$ could still be observed.

Further, Figure 2 shows that probability layer produces a higher accuracy than transfer learning. Following the published practice (Doersch et al. (2015); Han et al. (2016); Oquab et al. (2014); Shen et al. (2017); Yosinski et al. (2014)), all fully-connected layers after convolutional layers are finetuned on the generated dataset with same class distribution as the testing dataset for $5$ epochs. For all selected class numbers, probability layer performs better than transfer learning. This advantage of probability layer over transfer learning increases as the number of classes increase. We contribute this phenomenon to the fact that transfer learning may destroy the co-adaption between layers and deteriorate the performance on prediction, as reported in (Yosinski et al. (2014)). Another point worth noting is that when the number of classes increases over $90$, transfer learning would bring worse accuracy than the original model. In contrast, the probability layer can still bring $2\%$ advantage over the original model. We believe the reason is that the deterioration of co-adaption between layers leads to a decrease in accuracy and the reduction in the number of classes cannot make up this deterioration when the number of classes is $90$, which is almost same as the original class numbers. Probability layer does not need finetuning and thus avoid this problem. All these observations indicate that probability layer has better ability in using various environment than transfer learning.

Besides accuracy, we should also note that transfer learning during runtime consumes lots of energy. In each epoch, hundreds of images need to be processed by the mobile platform. As reported in existing works (Shen et al. (2017); Han et al. (2016)), a 14-second or even several-minute latency is required for runtime model adaption using transfer learning. By replacing transfer learning with probability layer, the model can be adapted during runtime without any overhead.

**Weak class skew ($p < 1$) with various percentage p.** A possible scenario is that $n$ classes only occupy the majority in the input stream while some other classes may also appear. Figure 3 shows that probability layer can still have good performance under this scenario. When $10$ classes occupy $90\%$ in the class skew, the accuracy with probability layer can get $87\%$, where a benefit of $15\%$ can be achieved over the original model. As the weight of the $5$ classes decreases, the benefit of probability layer also decreases, since the class skew becomes weaker. However, even if the $5$ classes only occupy $50\%$, a benefit of $5\%$ can still be achieved with probability layer. Similar results can also be observed for other numbers of classes, *e.g.*, $5$ classes. All these results show that probability layer can bring benefit in various class skew.

## 5.2 HIGH CONFIDENCE, HIGH ACCURACY

We justify our design of threshold in probability layer by exhibiting the correlation between high confidence and high accuracy, as well as that a large portion of images can get a prediction with high confidence. The phenomenon appears in various combinations of architectures and datasets. As an example, we show the analysis of DenseNet on CIFAR-100, see Figure 4.

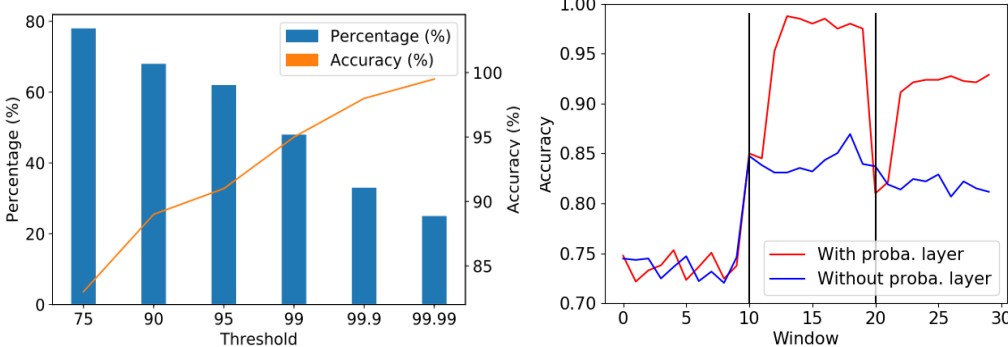

Figure 4: High confidence, high accuracy          Figure 5: Detection of class skews

High confidence leads to high accuracy, *i.e.*, the model maintains a higher accuracy on images that the model provides a high predicted probability. Specifically, for DenseNet with a top-1 accuracy of 73% on all CIFAR-100 testing images, the accuracy increases to 83% when we set the threshold $\omega$ to be 75%. When we increase further the threshold $\omega$ to be 99%, the accuracy would increase dramatically to 95.01%. Thus when a model has strong confidence in its prediction, we should better believe in the model instead of rescaling, *i.e.*, modifying the prediction based on environment through probability layer. In fact, the prediction may indicate the switching of class skew if several images are predicted with high confidence to be a class outside the current class skew.

High predicted probability can be obstained by a large portion of images, which justify the effectiveness of our design of threshold in probability layer. For instance, there are more than 60% images get predicted probability higher than 95%. For these images that the original model has strong confidence in its prediction, the probability layer will not interfere with the decision. The probability layer will step in when the original model is not sure and give suggestion to the probability layer according to the environment information. Through a series of experiments, we found that the threshold $\omega$ between 0.75 and 0.95 exhibits a similar performance. Thus we will use 0.9 as the value of threshold in the following section.

## 5.3 PROBABILITY LAYER ON DYNAMIC CLASS SKEW

In this section, we show that, under dynamic class skew, *i.e.*, class skews switches frequently in both class number and types, combining CSD and probability layer can efficiently detect class skew and achieve a similar performance as the case under static class skew, see Figure 5. Three phases reside with different number of classes, as 100, 5, and 10. Each time step is composed by 400 images composed by the corresponding number of classes with equal weights.

We can observe that, after the class skew switches, CSD can detect the class skew and probability layer can utilize the class skew to achieve a high accuracy. Specifically, the accuracy increases to 97% and 92% when the number of classes is 5 and 10, respectively. When the class skew switches, as indicated by time step 10 and 20, the CSD will realize that the class skew has switched immediately and starts to identify the new class skew. The new class skew will be forwarded to the probability layer, which will provide benefit similar to the case under static class skew.

## 5.4 PROBABILITY LAYER ON IMAGENET

In this section, we will show the performance of probability layer on ImageNet and other classification models, see Table 1. For the class skew with 10 classes and 50 classes, we randomly select corresponding number of classes for 100 times and exhibits the average results here. We can see that an average benefit of 25.32% and 19.98% in accuracy, over all four models on ImageNet, will be achieved on 10 and 50 class skews, respectively, which is similar to the results in section 5.1.

Table 1: Benefit of probability layer (PL) on ImageNet

| Model | Without PL | With PL | |
| --- | --- | --- | --- |
| | | 10 classes | 50 classes |
| Inception-V1 (Szegedy et al. (2015)) | 69.8 | 97.42 | 89.96 |
| MobileNet-V1 (Howard et al. (2017)) | 70.9 | 96.14 | 91.26 |
| VGG-16 (Simonyan & Zisserman (2014)) | 71.5 | 97.62 | 91.62 |
| ResNet-V1-50 (He et al. (2016)) | 75.2 | 97.5 | 94.5 |

## 5.5 BOTH INCREASING ACCURACY AND DECREASING RESOURCE CONSUMPTION

A promising way to achieve high speed up and accuracy is to combine acceleration methods with probability layer. For this to succeed, the acceleration methods should utilize different types of redundancy in the network. In this section, we verify that probability layer can be combined with an acceleration method of using spatial redundancy, *PerforatedCNN* (Figurnov et al. (2016)), to achieve high speed up while increasing top-1 accuracy.

We reimplemented the PerforatedCNN (Figurnov et al. (2016)) on DenseNet. PerforatedCNN makes use of spatial redundancy in the image by skipping evaluation in some of the spatial positions. Different from other methods in using spatial redundancy, i.e., increasing strides, PerforatedCNN will interpolate these skipped positions using nearest neighborhood, such that the output size will be unchanged. In this way, the architecture remains same and no finetuning is needed. The shortage of PerforatedCNN is that it may introduce a huge decrease in accuracy. Our experiments show that this drawback of PerforatedCNN could be made up by probability layer. Thus combining probability layer with other acceleration methods can both decrease computation and increase accuracy.

We first apply the probability layer to the network. Then we apply the spatial redundancy elimination methods to this network. In the whole process, no finetuning is needed. The PerforatedCNN is tested at the theoretical speedup level of 2x. The testing dataset contains 5 randomly selected classes with equal frequency. The results are presented in the table 2. Due to class effect, the original model will give a top-1 accuracy of 67.9%, which is slightly lower than the average accuracy of DenseNet on CIFAR-100. With the probability layer, the model without finetuning can increase the top-1 accuracy dramatically to be 98.4%. The PerforatedCNN will give a top-1 accuracy of 48.19% if we choose the theoretical speedup level of 2x, which is similar to the results reported in PerforatedCNN (Figurnov et al. (2016)). Adding the proposed method, the PerforatedCNN can give a top-1 accuracy of 92.20% while decreasing computation by half, which shows that probability layer complements spatial redundancy elimination methods perfectly and provides a promising perspective of combining probability layer with other acceleration methods.

Table 2: Summary

| Method | Mult. ↓ | Top-1 Accuracy |
| --- | --- | --- |
| Original Model | 1.0x | 67.79% |
| Probability Layer | 1.0x | 98.4% |
| Perforation | 2.0x | 48.19% |
| Combined Method | 2.0x | 92.20% |

## 6 CONCLUSION AND FUTURE WORK

We have presented probability layer which exploits runtime environment information to increase prediction accuracy. With no overhead, probability layer can achieve an equivalent or better performance than transfer learning. An effective class skew detector is provided, serving as the base for probability layer. Further, combining probability layer with existing pruning approaches both increases accuracy and decreases resource consumption. In the future, we will generalize our approach to other tasks, *i.e.*, RCNN for object detection and semantic segmentation, which would be straightforward as long as parts of the architecture is a CNN model.

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
