# OpenReview forum: "PCNN: Environment Adaptive Model Without Finetuning"
_ICLR.cc/2019/Conference_

### Official Review · AnonReviewer3 · 2018-10-31

**Rating:** 4
**Confidence:** 4

**Review:**

The idea proposed in this paper is to improve classification accuracy by making use of the context.
E.g. on the north pole we will see polar bears but no penguins, on Antartica we have no polar bears but many penguins.
Hence, if we apply our imagenet-like classifier in the wild, we can improve accuracy by taking into account changes in the prior distribution.

The paper proposes a way to rescale the probabilities to do exactly this and reports improved results on modified versions of
 CIFAR 10 and imagenet with artificial class skew. To achieve this, an additional trick is introduced where the re-scaling is only used when the model is not very certain of its prediction. And additional motivation for this work is that less compute resources are needed if the problem is simplified by utilizing class skew.

The core idea of the paper is interesting. However, I am not able to understand what exactly is done and I am 100% confident I cannot re-implement it. The authors already improved upon this in our interactions prior to the review deadline.
An additional issue is that the paper does not have a good baseline.
I would not like to dismiss the approach based on its simplicity. An elegant solution is always preferred. However, all the tasks are quite artificial and this limits the "impact" of this work. If an "natural" application/evaluation where this approach would be possible, it would strengthen the paper greatly.

For the reasons above I recommend rejection of the manuscript in the current state but I am confident that many of these issues can be resolved easily and if this is done I will update the review.

Missing information
----------------------------
- The original manuscript had a lot of information missing, but much of it has since been provided by the authors.
- In the static class skew experiment, were two passes over the data needed? Or was the Pt(i) pre-set? Would it also be possible to give details about LR, optimizer, LR schedule, batch size, .... for the transfer learning experiments. This would enhance reproducibility.
- For the imagenet experiments how was Pt(i) set in the if I assume correctly, static setting.

Possible additional baselines:
-----------------------------------------

We could make a simpler rescaling by changing the prior distribution and assuming everything else remains constant.
While this is a simplifying assumption, it is very easy to implement and should take only a couple of minutes to run.
P(i|x)=1/P(X)*P(X|i)*P(i)
Pt(i|x)=P(i|x)*Pt(i)/P(i)

One could also introduce another baseline where only the most probably classes are considered. Since this approach is clearly sub-optimal since it guarantees some mis-predictions it should serve as a lower bound on the performance that is to be expected.

---

### Official Review · AnonReviewer2 · 2018-11-02
**No technical contribution, Heuristic solution**

**Rating:** 3
**Confidence:** 4

**Review:**

This paper proposed a way to detect a skew in the distribution of classes in a stream of images and reweight the class priors accordingly, to estimate the final posterior probabilities of present classes. This probability re-calibration is referred to as the probability layer. A simple algorithm is proposed to detect the class distribution skew. The proposed benefit of this method is that they do not require fine-tuning any network parameters using newly skewed data.

Overall the method is quite simple and heuristic. The technical contribution - i) updating class priors online ii) detecting class skews, is marginal.

The evaluation is performed on a contrived setting of skewed imagenet images. I would have liked to see some evaluation on video stream data where the skews are more natural.

In real scenarios, the class specific appearances P_{X|Y}(x|i) as well as class distributions P_Y(i) change online. The method seems incapable to handle such problems.  In these situations, there is no simple fix, and one needs to resort to transfer.

---

### Official Review · AnonReviewer1 · 2018-11-05
**Simple Idea, Good Results But Novelty? Detailed Analysis?**

**Rating:** 4
**Confidence:** 4

**Review:**

The paper proposes a simple idea to calibrate probabilities outputted by a CNN model to adapt easily to environments where class distributions change with space and time (and are often skewed). The paper shows that such a simple approach is sufficient to get good accuracies without requiring any costly retraining or transfer learning. Thereby proving to give benefits in terms of resource consumption and at the same time giving better results than the state of the art.

However,
A] The proposed calibration doesn't take any CNN specific details into consideration, rather it is a general calibration method which was also proposed in Saerens et. al, 2002 (cited in the paper). It is unclear why the paper specifically talks about CNN.
B] The proposed Class Skew Detector is a simple method. Change-point detection is a well-studied area. The paper lacks a literature review in this area and a reasoning of why the proposed approach is preferred. Also, an independent analysis of how the class skew detector behaves in the face of rapidly changing class skews versus slow changing class skews is warranted here. Particularly, given that the paper proposes to use this approaches in mobile which may work in both rapid and slow changing class skews.
C] The Class Skew Detector is dependent on the base model. Thus, it is also likely that the empirical distribution estimated is biased and yet the final accuracies reported are much higher than the base model accuracies. There is something interesting happening here. An analysis of the robustness of the proposed approach in the face of noisy class skew detection could potentially make this paper a stronger work.
D] The analysis in the paper has largely focused on pre-trained models. However, another analysis that could have been useful here is, varying the quality of the classifier (e.g. classifier trained on skewed training data vs. balanced training data) and measuring how the quality of the classifier correlates with the final performance. Maybe even attempt to answer the question "which classifiers are likely to work with this approach?" In fact, this analysis can be either done in a general context of any classifier or just CNN's and identifying whether certain properties of CNN help in getting better performance.

The paper lacks novelty and at the same time, it is not quite compensating that with a detailed analysis of the work. The problem is interesting and I like the work because the approach is simple and the results look good. I think with a stronger focus on more detailed analysis, this can be a good submission to an applied conference like MobiCom etc.

By the way, the paper is riddled with several spelling errors -
"filed" -> "field", page 1, second paragraph, last line
"complimentary" -> "complementary", page 2, section 2, paragraph 1, last line
"epoches" -> "epochs", page 2, section 2, transfer learning, second paragraph, second last line
"CNNs does not use" -> "CNNs do not use", page 3, section 3, intuition, first paragraph, first line
"formular" -> "formula", page 4, above equation 4
Equation 4 has a typo in the denominator, P_t(i) should be P_t(j), same with Equation 5
"obstained" -> "obtained", page 7, second paragraph, first line
"adaptation" is almost everywhere spelled as "adaption"

---

### Comment · AnonReviewer3 · 2018-10-31
**Request for clarification**

Dear Authors,

Could you please clarify the transition from equation 3 to equation 4. I do not understand how this step is made.

It would be helpful if you could clarify this before I submit the official review.

---

> ### Author Response · Authors · 2018-11-01
> **Clarification on the transition from euqation 3 to equation 4**
>
> Dear reviewer,
>
> Thanks for your comment! There is a small typo in equation 4, which has no influence over other parts in our paper. (This clarification could be better read in pdf version (https://drive.google.com/file/d/1M1t0CjZWmcolfELb-kkqKVWtcqR9A6mg/view?usp=sharing) due to latex equations.)
>
> Instead of
> \begin{equation*}
>     P_t(i|X) = \frac{\frac{P_t(i)}{P(i)} \cdot P(i|X)}{\sum_{j=1}^n \frac{P_t(i)}{P(j)} \cdot P(j|X)}
> \end{equation*},
> it should be
> \begin{align*}
>      P_t(i|X)  = \frac{\frac{P_t(i)}{P(i)} \cdot P(i|X)}{\sum_{j=1}^n \frac{P_t(j)}{P(j)} \cdot P(j|X)}
> \end{align*}.
> Note the single $i$ in the denominator has been replaced by $j$.
>
> The following is the detailed proof on the transition from equation 3 to equation 4.
>
> In equation 3, we have $P_t(i|X) = \frac{P_t(i)}{P(i)} \cdot \frac{P(X)}{P_t(X)} \cdot P(i|X)$. We also have $\sum_{i=1}^n P_t(i|X) = 1$, based on the property of probability. Together, we can find that
> \begin{align*}
>     1 & = \sum_{i=1}^n P_t(i|X) \\
>       & = \sum_{i=1}^n\frac{P_t(i)}{P(i)} \cdot \frac{P(X)}{P_t(X)} \cdot P(i|X) \\
>       & = \frac{P(X)}{P_t(X)} \cdot \sum_{i=1}^n \frac{P_t(i)}{P(i)} \cdot P(i|X)
> \end{align*}
> The second equality holds by using equation 3. The third equality holds since $\frac{P(X)}{P_t(X)}$ does not change over $i$.
>
> Thus, we can have
> \begin{align*}
>     \frac{P(X)}{P_t(X)} & = \frac{1}{\sum_{i=1}^n \frac{P_t(i)}{P(i)} \cdot P(i|X)}   \\
>                         & = \frac{1}{\sum_{j=1}^n \frac{P_t(j)}{P(j)} \cdot P(j|X)}
> \end{align*}
> The second equality holds since every $i$ has been replaced with $j$. We conduct this replacement to avoid confusement with $i$ used in equation 3 and equation 4.
>
> Use this equation to replace $\frac{P(X)}{P_t(X)}$ in equation 3, we can get
> \begin{align*}
>     P_t(i|X) & =  \frac{P_t(i)}{P(i)} \cdot P(i|X) \cdot \frac{P(X)}{P_t(X)} \\
>              & = \frac{P_t(i)}{P(i)} \cdot P(i|X) \cdot \frac{1}{\sum_{j=1}^n \frac{P_t(j)}{P(j)} \cdot P(j|X)} \\
>              & = \frac{\frac{P_t(i)}{P(i)} \cdot P(i|X)}{\sum_{j=1}^n \frac{P_t(j)}{P(j)} \cdot P(j|X)}
> \end{align*}

---

> > ### Comment · AnonReviewer3 · 2018-11-01
> > **Thanks for this response**
> >
> > Thanks for this response,
> >
> > Can you also expand on section 4. The notation used in algorithm 1 is not detailed.
> > What is of particular interest is how Pt(i) is computed.

---

> > > ### Author Response · Authors · 2018-11-01
> > > **Clarification on section 4.**
> > >
> > > Dear reviewer:
> > >
> > > Thanks for your comment!  (This clarification could be better read in pdf version (https://drive.google.com/file/d/17wFjCrhnNjcoIeV5v537gvcw9bH3KekX/view?usp=sharing) due to latex equations.)
> > >
> > >
> > > We estimate $P_t(i)$ with the \textit{empirical class distribution} [1] in a short time window (every $\omega_{min}$). In algorithm 1, $y_t$ indicates the prediction result for the $t$-th input frame classified by the model $h(\cdot)$. $S_j$ indicates the empirical distribution in the $j$-th time window (the utilization of time window will be justified in \textbf{Assumption} paragraph) and $\oplus$ indicates the concatenation of two distributions. $S_j \leftarrow S_j \oplus [y_t]$ means that every new prediction result $y_t$ will be incorporated into the \textit{empirical class distribution} $S_j$ composed by all $\omega_{min}$ predictions, which will be computed as following:
> > > \begin{equation}
> > >   S_j(i) = \frac{1}{\omega_{min}} \sum_{t=1}^{\omega_{min}}\mathbbm{1}_{y_t \leq i}
> > > \end{equation}
> > > . The $P_t(i)$ can be derived from empirical class distribution $S_j(i)$ by
> > > \begin{equation}
> > >     P_t(i) = S_j(i) - S_j(i-1)
> > > \end{equation}
> > >
> > >
> > > The if statement of $|| S_{j-1}, S_j|| \leq \pi_r$ is proposed for detecting the switch of class skew, as detailed in the following.
> > >
> > >
> > > \paragraph{Assumption.}As described in the first paragraph of section 4, the only assumption we hold is that the class skew in a scenario remains unchanged. Formally, let assume the existence of a partition (scenario for a class skew) $\pi: N^+ \rightarrow N^+$ over the input stream, where $\pi(t)$ refers to the class skew that $t$-th image belongs to. Here, each partition maintains a distribution $T_{\pi(t)}$ and the image $(x_t, y_t)$ is drawn randomly (\textit{i.i.d.}) from distribution $T_{\pi(t)}$. Here, the overall series is composed by a sequence of abruptly-changing partitions and the distribution within each partition remains same. This is a very weak but realistic assumption, since we do not have any other assumptions on how long a stationary distribution exists. Thus our proposed algorithm needs to not only detect the underlying distribution $T_{\pi(t)}$ ($P_t(i)$ is the probability of each class $i$ in the distribution $T_{\pi(t)}$), but also recognize the start time and end time for each partition $\pi(t)$ (class skew) in an untrimmed streams of data.
> > >
> > > \paragraph{Proposed approach.}As described in the second paragraph of section 4, we propose a windowed class skew detector to approximate the underlying distribution, as well as the start time and end time for each partition $\pi(t)$ (class skew). Here, the empirical distribution $S_j$ in each window $j$ can be obtained to estimate the $P_t(i)$. Furher, the start time and end time of each partition $\pi(t)$ (class skew) can be decided when there is a dramatic change in empirical class distributions $S_{j-1}$ and $S_{j}$ from adjacent windows $j-1$ and $j$. A dramatic change is decided when
> > > \begin{equation}
> > >   \underset{i}{\text{sup}} | S_{j}(i) - S_{j-1}(i) | \geq \frac{\pi_r}{\omega_{min}}
> > > \end{equation}
> > > , where $\omega_{min} = 30$ and $\pi_r = 2$ in our evaluations.
> > >
> > > \paragraph{Edge case when class skew switches.}Our proposed probability layer can handle the edge case, \textit{i.e.}, a small turbulence to class skew has happened. For example, $10$ people stays in a lab and a stranger suddenly visits. This edge case is handled by the weak class skew (p<1) in our evaluation section.
> > >
> > > We apologize for not providing enough detail for the Algorithm 1. We will revise it in our final version.
> > >
> > > \begin{wrapfigure}{R}{0.35\textwidth}
> > >     \begin{minipage}{0.35\textwidth}
> > >       \begin{algorithm}[H]
> > >         \caption{CSD algorithm}
> > >         \begin{algorithmic}
> > >             \Function{CSD}{$ $} \label{alg: WEG}
> > >                 \For{$t$ in $1, ..., w_{min}$}
> > >                     \State $y_t \leftarrow h(t)$
> > >                     \State $S_j \leftarrow S_j \oplus [y_t]$
> > >                 \EndFor
> > >                 \If{$|| S_{j-1}, S_j|| \leq \pi_r$}
> > >                     \State $S_j \leftarrow S_{j-1} \oplus S_j$
> > >                 \EndIf
> > >                 \State \Return $S_j$
> > >             \EndFunction
> > >         \end{algorithmic}
> > >          \label{alg: algorithm}
> > >       \end{algorithm}
> > >     \end{minipage}
> > > \end{wrapfigure}
> > >
> > > [1]  J. Shao.Mathematical Statistics. Springer Texts in Statistics. Springer, 2003.

---

### Meta-Review · Area_Chair1 · 2018-12-10
**metareview: no rebuttal**

**Confidence:** 5
**Recommendation:** Reject

**Metareview:**

All reviewers rate the paper as below threshold. While the authors responded to an earlier request for clarification, there is no rebuttal to the actual reviews. Thus, there is no basis by which the paper can be accepted.